# Anthropometric Status, Diet, and Dental Caries among Schoolchildren

**DOI:** 10.3390/ijerph18137027

**Published:** 2021-06-30

**Authors:** Chin-En Yen, Yuh-Yih Lin, Suh-Woan Hu

**Affiliations:** 1Department of Early Childhood Development and Education, Chaoyang University of Technology, Taichung 413, Taiwan; ceyen@cyut.edu.tw; 2Department of Dentistry, Chung Shan Medical University, Taichung 402, Taiwan; yuhyih@csmu.edu.tw; 3Institute of Oral Sciences, Chung Shan Medical University, Taichung 402, Taiwan; 4Department of Dentistry, Chung Shan Medical University Hospital, Taichung 402, Taiwan

**Keywords:** body mass index, dental caries, obesity, overweight, children

## Abstract

Childhood dental caries and obesity are prevalent health problems. Results from previous studies of the caries–obesity relationship are conflicting. This study aimed to assess the association between anthropometric status and dental caries among schoolchildren, taking into account dietary habits, oral hygiene, and sociodemographic factors. This cross-sectional study recruited 569 children aged 6–12 years from five elementary schools in central Taiwan. Each child underwent an oral health examination and anthropometric measurements. The DMFT (decayed, missing due to caries, and filled permanent teeth) and deft (decayed, extracted, and filled primary teeth) indexes were calculated to record caries experience. A structured questionnaire was used to collect information on food intake frequency and other related factors. The World Health Organization’s reference data was applied to define weight status: obese, overweight, and normal/underweight. The results showed that the mean (±standard deviation) deft and DMFT scores were 2.3 ± 2.6 and 0.7 ± 1.2, respectively, among participating children. The prevalence of obesity and overweight was 18.1% and 18.5%, respectively. After comprehensive evaluation of potential confounders, weight status was not an independent predictor of DMFT or deft scores in the negative binomial regression models. In conclusion, weight status was not associated with caries scores in primary or permanent teeth among 6–12 year-old schoolchildren.

## 1. Introduction

Childhood dental caries is one of the major public health issues in numerous countries. The Global Burden of Disease 2017 Study estimated that globally 532 million children had untreated caries in primary teeth [1]. The World Health Organization (WHO) reported that 60–90% of school-aged children experienced dental caries [2]. Dental caries may cause pain, impair masticatory function, and affect growth, general health, and quality of life [2,3,4,5].

Overweight/obesity is one of the most common health problems that affect children worldwide. Globally, about 18% of children and adolescents aged 5–19 years were overweight or obese in 2016 [6]. Childhood obesity can have profound effects on children’s physical health, psychosocial health and quality of life [7,8,9,10]. Furthermore, overweight/obese children are more likely to become obese adults and have increased risk of multiple diseases, such as cardiovascular diseases and diabetes, in adulthood [7,8,9,10].

The relationship between dental caries and childhood overweight/obesity has been examined among school-aged children in many studies; however, the findings are inconsistent. Overweight/obese children have been found to have significantly higher caries scores [11,12] than those of normal weight. Conversely, some studies reported that overweight/obese children had lower caries scores [13,14,15] or were less likely to have dental caries [14,16,17,18,19,20,21] compared to children of normal weight. Furthermore, results from several studies indicated that there was no significant association between weight status and caries scores [22,23,24,25] or caries prevalence [22,24,25,26]. Several systematic reviews of the childhood obesity-caries relationship indicated that various confounding factors were not evaluated in several previous studies and might affect the findings [27,28,29,30]. Factors reported to be associated with obesity [7,8,9,31,32] or with dental caries [3,4], such as socio-demographic status, dietary habits, physical activity, oral hygiene habits, medical history and fluoride exposure, need to be considered in the analysis.

Childhood dental caries and obesity have become important public health problems in Taiwan. A nationwide survey in 2013 reported a caries experience prevalence of 67.5% in primary teeth and 50.0% in permanent teeth among 6–12 year-old schoolchildren in Taiwan [33]. Yet, a national survey in the United States during 2011–2012 found that only 21.3% of children aged 6–11 years experienced caries in permanent teeth [34]. According to the 2013–2016 nationwide survey in Taiwan, 26.7% of elementary schoolchildren aged 7–12 years were overweight (11.1%) or obese (15.6%) [35]. The prevalence of obesity has doubled in 10 years (only 7.0% in the 1993–1996 national survey) [35]. However, the relationship between dental caries and anthropometric status among schoolchildren in Taiwan is not well understood. Furthermore, the discrepancies in previous findings merit further investigation, and studies with comprehensive evaluation of potential confounders may help to clarify the childhood obesity–caries relationship. This study aimed to assess the association between anthropometric status, diet, and dental caries among schoolchildren.

## 2. Materials and Methods

### 2.1. Study Design and Participants

This cross-sectional study was conducted in Taichung, Taiwan, between October 2019 and May 2020. Taichung city has about 2.8 million citizens and consists of 29 administrative districts, which are located in the mountain, basin, plain, or coastal areas. The city has a dentist-to-population ratio of 69.7 per 100,000, higher than 64.1 per 100,000 in Taiwan. In order to select schools from areas of varied urbanization levels, all 240 elementary schools in the city were categorized into three groups according to the location, the main drinking water supply, and the urbanization stratification [36] of their districts. Two schools were randomly sampled from each group with the probability proportional to size. Five schools agreed to participate in this study. Within each school, we invited two classes from each of the six grades to participate in the study. Invitation letters and consent forms were sent to 1380 students from these classes. In total, 569 children completed the questionnaire, the oral examination, and the anthropometric assessment, and were included for subsequent data analysis. Age of participants ranged from 6.4 to 12.7 years, with a mean (±standard deviation, SD) of 9.4 ± 1.4 years. The percentages for boys and girls were 49.7% and 50.3%, respectively (Table 1). A written informed consent was obtained from each child and her/his parents or guardians. The study protocol was approved by the Institutional Review Board of Chung Shan Medical University Hospital (CSMUH No: CS18238) in Taichung, Taiwan.

### 2.2. Questionnaire Survey

A structured questionnaire was derived from previous studies [37,38], and it was updated, reviewed by experts, and pilot-tested. Each participating child’s parents or guardians answered the questionnaire with the following information: (1) demographics, including age, gender, and parents’ educational levels; (2) physical activity level and medical history for physician-diagnosed disorders; (3) frequency of dental visits; (4) oral hygiene behavior; and (5) usage of fluoride products, including fluoride toothpaste, varnish, mouthrinse, and fluoridated salt. A food frequency questionnaire was used to assess each child’s frequency of consumption of 25 food groups.

### 2.3. Oral Health Examinations

Each child’s dental health condition was assessed following the WHO’s Oral Health Surveys-Basic Methods [39] guidelines. Two trained and calibrated dentists performed the oral examination for all the participants. The calibration practice, in which 30 schoolchildren of different ages were examined repeatedly by two dentists, yielded a kappa coefficient of 0.84 for the inter-examiner reliability on caries diagnosis. For each child, the DMFT (decayed, missing due to caries, and filled permanent teeth) and deft (decayed, indicated for extraction, and filled primary teeth) indexes were calculated and used to record caries experience. The number of erupted primary teeth and permanent teeth were also recorded. Children with DMFT score ≥ 1 were defined as having caries experience in permanent teeth, and those with deft score ≥ 1 were classified as having caries experience in primary teeth.

### 2.4. Anthropometric Measurements

Each child’s anthropometric status was measured by the same trained researcher. Body weight and height were measured in light clothing and without shoes. The formula for calculating BMI was weight (kg)/height (m) ^2^. Body fat (%) was assessed using a body fat meter (Tanita BF049, Tokyo, Japan). Mid-arm-circumference was measured with tape at the mid-point of the upper dominant arm around the circumference on a bare arm. Triceps skinfold thickness was measured at the midpoint of the dominant upper posterior arm using a Lange skinfold caliper (Cambridge, UK). The age- and sex-standardized BMI-z-score was calculated using the World Health Organization’s 2007 growth reference data for 5–19 years [40,41]. According to the WHO cut-off points [42], each child’s weight status was defined as (1) “obese”: with BMI-z-score > 2 SDs, (2) “overweight”: BMI-z-score > 1 to ≤2 SDs, (3) “normal weight”: BMI-z-score ≥ −2 to ≤ 1 SD, and (4) “underweight”: BMI-z-score < −2 SDs. Since the “underweight” group had a very small sample size (*n* = 7), it was combined with the “normal weight” group to form the “normal/underweight” group (BMI-z-score ≤ 1 SD) for further data analysis.

### 2.5. Data Analysis

The data analysis started with descriptive statistics for all important factors. The data distributions of continuous variables were checked using the histogram and box plots, and the Kolmogorov–Smirnov test was applied to test for the normality of the distributions. Since DMFT and deft scores were not normally distributed, the non-parametric methods were applied for the unadjusted comparisons of caries scores. The Wilcoxon rank-sum test was used to compare caries scores between groups. The Kruskal–Wallis test was used for comparisons of caries scores and tooth numbers among groups, and if significant, followed by Dunn’s multiple comparison procedure. Spearman rank correlation was used to evaluate the correlations between caries scores and anthropometric measures. The chi-squared test was used to assess the relationship between categorical variables, such as caries prevalence, sociodemographic factors, and weight status. Variables significantly associated with both caries experience and weight status were considered as confounders in this study and were included for subsequent regression analysis. Finally, the negative binomial regression analysis was implemented to evaluate the association between weight status and caries scores, adjusting for confounders identified in the bivariate analyses. The SAS version 9.4 software (SAS Institute Inc., Cary, NC, USA) was applied for data analyses. An alpha level of 0.05 was used for all statistical tests.

## 3. Results

### 3.1. Descriptive Characteristics and Dental Caries Experience of Participants

The descriptive characteristics and dental caries experiences of 569 participating children are shown on Table 1. Factors significantly associated with caries experience in primary teeth were age group, father’s education, dental visit in 6 months, and participation in the school fluoride mouthrinse program. Factors significantly related to caries experience in permanent teeth included gender, age group, father’s education, and mother’s education level. Fluoride tablets and fluoridated salt were used by 12.8% and 4.4% of participants, respectively, and were not significantly related to caries experience (data not shown on table). Regarding the child’s medical history, the prevalence of the top five physician-diagnosed diseases among participating children were: allergic rhinitis (20.7%), otitis media (6.4%), atopic dermatitis (5.4%), asthma (3.5%), and sinusitis (3.3%) (data not shown on table). The distribution of allergic rhinitis and other diseases were not significantly associated with caries experience.

### 3.2. Food Intake Frequency and Dental Caries

We analyzed the associations between dental caries and the intake frequency of 25 food groups. Table 2 presents the relationship of seven food groups and caries scores. Children with more frequent intake of sweet snacks, including fruit juice, Yakult and other yogurt drinks, candy and chocolate, cake and cookies, and sugary drinks, had significantly higher deft scores. Intake frequency of other foods, such as meat, dairy products, fruits and vegetables, tea, and dietary supplements, were not significantly associated with caries in primary or permanent teeth (data not shown on table).

### 3.3. Correlation of Anthropometric Measures and Caries Scores

The mean ± SD (median, range) of BMI and BMI-z-score was 18.4 ± 3.7 kg/m^2^ (17.2, 12.4–32.4) and 0.7 ± 1.3 (0.4, −3.7–4.8), respectively, for all children. Anthropometric measures were significantly negatively correlated with deft scores, but positively correlated with DMFT scores and age (Table 3). BMI-z-score was not correlated with caries scores, but was significantly correlated with anthropometric measures. Numbers of erupted primary or permanent teeth, respectively, were significantly correlated with age, BMI-z-score, deft, and DMFT score.

### 3.4. Relationship of Weight Status with Important Factors

Table 4 shows the relationship of weight status with sociodemographic factors, caries experience, medical history, and food intake frequency. According to the age- and gender-specific BMI standard of WHO, 36.6% of the children were either overweight (18.5%) or obese (18.1%). Obese children had significantly lower deft score and lower number of primary teeth than normal/underweight children. Other factors significantly associated with weight status were gender, father’s education, mother’s education, frequency of tooth-brushing, and physical activity level. Frequency of intake of fast food, sweet snacks, and sugared drinks were not different among weight status groups. Fluoride usage variables were not significantly associated with weight status (data not shown on table).

### 3.5. Adjusted Associations between Weight Status and Caries Index Scores

Results from the negative binomial regression analyses indicated that weight status was not significantly associated with either DMFT or deft scores, after adjusting for confounding factors identified in the bivariate analyses (Table 5). Because age and number of teeth were highly correlated and multicollinearity was detected when both factors were included in the same regression model, these two factors were adjusted in separate models. Compared to the model with adjustment for age, the model with number of teeth had better fit and, therefore, was considered the final model. In the sensitivity analysis, exclusion of seven underweight children from the negative binomial regression models generated the same results. Moreover, BMI-z-score was not significantly associated with DMFT or deft scores in the negative binomial regression analyses.

## 4. Discussion

In the present study, there was no significant association between weight status and deft or DMFT scores among 6–12 year-old schoolchildren, after adjusting for confounding factors. These study results are consistent with previous studies that found no association between weight status and caries scores of primary or permanent teeth among 6–12 year-old children in India [25], 5–8 year-old children in the Netherlands [24], 8–12 year-old schoolchildren in Mexico [43], or 6, 12, and 15 year-old children in Spain [23]. However, our study results conflicted with findings of some other investigations. Several studies reported an inverse association between overweight/obesity and deft scores among 8 year-old children in China [44], 5th graders in Kuwaiti [45], or 5–18 year-old children in India [14] as well as between overweight and DMFT scores among 6–12 year-old children in India [15]. Nonetheless, a study of 9–12 year-old children in Germany reported a positive association between overweight/obesity and DMFT scores [11].

One of the reasons for these discrepancies is that a variety of confounding factors were not considered by some studies mentioned above. Both childhood dental caries and obesity are multifactorial conditions and share some risk factors [27,29,30,46]. It is important to evaluate and control the effects of confounding factors in order to obtain an un-biased association. Secondly, previous studies have focused on different measures of caries experiences (such as caries prevalence of primary teeth, caries prevalence of permanent teeth, overall caries prevalence, deft, DMFT, or combining deft and DMFT), used different national or international reference standards for calculating BMI-z-score, classified weight status in different ways (such as four groups—obese, overweight, normal, underweight; three groups—obesity, overweight, normal/underweight; or two groups—overweight vs. not overweight), and investigated their relationship among children of various age groups. Therefore, direct comparisons of results from relevant studies are complicated. Lastly, previous studies assessed the association among children of different ethnicity/culture groups or geographical areas, and these factors might have modified the caries–obesity relationship. A study in central Italy showed that mother’s nationality was significantly associated with caries experience in primary dentition among preschool children [47].

In this study, we did not observe a significant relationship between weight status and dental caries among 6–12 year-old schoolchildren, after comprehensively evaluating the potential confounding effects of factors reported to be associated with childhood obesity, including socio-demographic status, dietary habits, consumption of sugary beverages, snacks or fast food, and physical activity [7,8,9,31], as well as the well-known risk factors or preventive measures for dental caries, such as age, gender, socio-economic status, oral hygiene, dietary habits, and fluoride usage [3,4]. In addition, the number of primary or permanent teeth, which was seldom considered in previous investigations, appeared to be correlated with caries scores and was also significantly different among weight status groups in this study. Children aged 6–12 years generally have mixed dentition. As the child’s age increases, her/his permanent teeth erupt and primary teeth exfoliate. Therefore, the number of primary teeth at risk for caries decreases, while the number of permanent teeth at risk for caries increases. Moreover, a previous study reported that overweight schoolchildren had more erupted permanent teeth than children in the lower BMI groups [48]. Inclusion of this confounding factor in the regression models of our study helped to clarify the relationship between obesity and dental caries. Furthermore, we have assessed the potential confounding effects of the child’s medical conditions, such as allergic rhinitis and asthma, which were reported to be positively or inversely associated with dental caries among children or adolescents [49,50,51,52] as well as related to obesity in children [53,54]. However, we could not rule out the possibility that the results of the present study have been biased by other confounders. Several factors related to childhood obesity, such as birthweight, breast feeding in infancy [8,55], long-term dietary habits, family function, and family history of obesity [8,31,46], were not evaluated in the analysis and could have confounded the observed results.

Although many risk factors or preventive factors related to caries or obesity were evaluated in the data analysis in order to clarify whether weight status was an independent predictor for caries scores, this study also has several limitations. Firstly, it was challenging to elucidate the temporal relationship between weight status and dental caries in a cross-sectional study. The child’s dietary intake habits might affect their BMI and caries experience, and vice versa. Associations observed in this study did not indicate a causal relationship. Long-term follow-up studies are needed to investigate the inter-relationship between childhood obesity and caries. Secondly, the WHO criteria were used to evaluate each child’s dental caries experience in this study. The caries activity status and the severity of carious lesions were not evaluated. Future studies may apply the International Caries Detection and Assessment System [56,57] to have a more comprehensive assessment of dental caries status of schoolchildren. Thirdly, about 41% of the invited children completed the questionnaire and oral health examination. The study participants might not be representative of schoolchildren in the study area. During the study period, the second semester of all elementary schools was postponed by the Taiwan government due to the COVID-19 pandemic. The project schedule was interrupted and we were not able to recruit more children to increase the response rate. Based on available information, the response rates were not significantly different between boys and girls. Finally, as aforementioned, several factors reported to be related to childhood obesity were not evaluated in the current study and might have affected the observed association.

## 5. Conclusions

There was no significant association between weight status and dental caries scores among 6–12 year-old schoolchildren, after comprehensive evaluation of potential confounders. Further research with longitudinal follow-up and more thorough assessment of potential confounding factors will better elucidate the complex relationship between childhood obesity and dental caries.

## Figures and Tables

**Table 1 ijerph-18-07027-t001:** Demographic characteristics, oral health habits, fluoride use, and dental caries experience of participating schoolchildren.

			Primary Teeth			Permanent Teeth	
Variables	*N*	Deft = 0 *n* (%)	Deft ≥ 1 *n* (%)	Deft Score Mean ± SD	DMFT = 0 *n* (%)	DMFT ≥ 1 *n* (%)	DMFT Score Mean ± SD
All children	569	210 (36.9)	359 (63.1)	2.3 ± 2.6	389 (68.4)	180 (31.6)	0.7 ± 1.2
Gender							
Boys	283	98 (34.6)	185 (65.4)	2.4 ± 2.7	206 (72.8)	77 (27.2) ^a,^*	0.6 ± 1.2 ^b,^*
Girls	286	112 (39.2)	174 (60.8)	2.2 ± 2.4	183 (64.0)	103 (36.0)	0.7 ± 1.2
Age group							
6–8 yrs	261	43 (16.5)	218 (83.5) ^a,^***	3.6 ± 2.8 ^b,^***	222 (85.1)	39 (14.9) ^a,^***	0.2 ± 0.6 ^b,^***
9–12 yrs	308	167 (54.2)	141 (45.8)	1.2 ± 1.7	167 (54.2)	141 (45.8)	1.0 ± 1.5
Area of school							
Urban	124	48 (38.7)	76 (61.3)	2.1 ± 2.3	78 (62.9)	46 (37.1)	0.8 ± 1.6
Sub-urban	213	89 (41.8)	124 (58.2)	2.1 ± 2.5	143 (67.1)	70 (32.9)	0.6 ± 1.1
Rural	232	73 (31.5)	159 (68.5)	2.6 ± 2.8	168 (72.4)	64 (27.6)	0.6 ± 1.1
Father’s education level							
Less than high school	64	17 (26.6)	47 (73.4)	2.9 ± 2.6 ^c,^*	35 (54.7)	29 (45.3) ^a,^*	1.1 ± 1.8 ^c,^*
High school	229	92 (40.2)	137 (59.8)	2.3 ± 2.8	156 (68.1)	73 (31.9)	0.7 ± 1.1
College and above	270	101 (37.4)	169 (62.6)	2.1 ± 2.4	195 (72.2)	75 (27.8)	0.5 ± 1.1
Mother’s education level							
Less than high school	51	16 (31.3)	35 (68.6)	2.9 ± 2.8	27 (52.9)	24 (47.1) ^a,^*	0.9 ± 1.3 ^c,^*
High school	193	72 (37.3)	121 (62.7)	2.4 ± 2.7	127 (65.8)	66 (34.2)	0.8 ± 1.4
College and above	313	116 (37.7)	197 (62.9)	2.1 ± 2.4	226 (72.2)	87 (27.8)	0.5 ± 1.1
Medical history and activity							
History of allergic rhinitis							
Yes	119	50 (42.0)	69 (58.0)	2.0 ± 2.6	80 (67.2)	39 (32.8)	0.8 ± 1.3
No	441	157 (35.6)	284 (64.4)	2.4 ± 2.6	302 (68.5)	139 (31.5)	0.6 ± 1.2
Physical activity							
High	149	54 (36.2)	95 (63.8)	2.5 ± 2.8	105 (70.5)	44 (29.5)	0.5 ± 0.9
Moderate	345	125 (36.2)	220 (63.8)	2.3 ± 2.5	233 (67.5)	112 (32.5)	0.7 ± 1.3
Low	73	29 (39.7)	44 (60.3)	2.0 ± 2.4	50 (68.5)	23 (31.5)	0.7 ± 1.3
Oral hygiene habit							
Frequency of tooth-brushing							
<=1 time/day	96	30 (31.3)	66 (68.7)	2.7 ± 2.8	73 (76.0)	23 (24.0)	0.6 ± 1.4
2 times/day	308	113 (36.7)	195 (63.3)	2.3 ± 2.5	213 (69.2)	95 (30.8)	0.6 ± 1.2
>=3 times/day	163	65 (39.9)	98 (60.1)	2.1 ± 2.6	103 (63.2)	60 (36.8)	0.7 ± 1.1
Floss daily							
Yes	332	126 (38.0)	206 (62.0)	2.2 ± 2.5	217 (65.4)	115 (34.6)	0.7 ± 1.1
No	234	82 (35.0)	152 (65.0)	2.6 ± 2.8	171 (73.1)	63 (26.9)	0.6 ± 1.3
Dental visit in 6 months							
Yes	368	109 (29.6)	259 (70.4) ^a,^***	2.6 ± 2.7 ^b,^***	254 (69.0)	114 (31.0)	0.6 ± 1.1
No	195	98 (50.3)	97 (49.7)	1.7 ± 2.4	131 (67.2)	64 (32.8)	0.7 ± 1.4
Fluoride usage							
Use fluoride toothpaste daily							
Yes	407	149 (36.6)	258 (63.4)	2.3 ± 2.5	280 (68.8)	127 (31.2)	0.6 ± 1.2
No	152	56 (36.8)	96 (63.2)	2.5 ± 2.9	105 (69.1)	47 (30.9)	0.6 ± 1.1
Used fluoride varnish							
Yes	517	188 (36.4)	329 (63.6)	2.3 ± 2.6	357 (69.1)	160 (30.9)	0.6 ± 1.1
No/don’t remember	48	19 (39.6)	29 (60.4)	2.3 ± 2.8	30 (62.5)	18 (37.5)	1.1 ± 2.1
Participated in school fluoride mouthrinse program							
Yes	449	176 (39.2)	273 (60.8) ^a,^*	2.1 ± 2.5 ^b,^**	304 (67.7)	145 (32.3)	0.6 ± 1.2
No	107	29 (27.1)	78 (72.9)	3.0 ± 2.9	78 (72.9)	29 (27.1)	0.6 ± 1.2

^a^ Chi-squared test for comparisons of caries prevalence. ^b^ Wilcoxon rank sum test for comparison of caries scores between groups. ^c^ Kruskal-Wallis test for comparison of caries scores among groups. **p* < 0.05, ** *p* < 0.01, ****p* < 0.001.

**Table 2 ijerph-18-07027-t002:** The relationship of food intake frequency and caries scores.

Food Items	*N*	Deft Score	DMFT Score
Fruit juice			
G1: <1 time/month	289	2.1 ± 2.5 ^a,^*	0.7 ± 1.3
G2: 1–3 times/month	127	2.3 ± 2.4	0.6 ± 1.1
G3: 1–6 times/week	91	3.2 ± 3.2	0.6 ± 1.2
G4: >=1 time/day	55	2.1 ± 2.6	0.6 ± 1.1
		#: G3 > G1	
Yakult & other yogurt drinks			
G1: <1 time/month	190	2.0 ± 2.5 **	0.8 ± 1.4
G2: 1–3 times/month	210	2.2 ± 2.5	0.6 ± 1.1
G3: 1–6 times/week	123	2.9 ± 2.7	0.6 ± 1.1
G4: >=1 time/day	42	2.3 ± 2.9	0.5 ± 1.0
		#: G3 > G1, 2	
Fast food			
G1: <1 time/month	145	2.4 ± 2.6	0.8 ± 1.3 *
G2: 1–3 times/month	265	2.3 ± 2.6	0.6 ± 1.1
G3: 1–6 times/week	133	2.2 ± 2.6	0.5 ± 1.0
G4: >=1 time/day	20	2.4 ± 2.6	1.4 ± 2.5
Candy & chocolate			
G1: <1 time/month	87	1.7 ± 2.1 *	1.0 ± 1.7 *
G2: 1–3 times/month	187	2.4 ± 2.9	0.6 ± 1.2
G3: 1–6 times/week	228	2.6 ± 2.5	0.5 ± 1.0
G4: >=1 time/day	62	2.0 ± 2.5	0.8 ± 1.3
		#: G3 > G1	#: G3 < G1
Cake, bread & cookies			
G1: <1 time/month	55	1.6 ± 2.7 *	1.0 ± 1.5
G2: 1–3 times/month	130	2.4 ± 2.5	0.6 ± 1.4
G3: 1–6 times/week	288	2.4 ± 2.5	0.6 ± 1.1
G4: >=1 time/day	92	2.5 ± 2.8	0.7 ± 1.1
		#: G3 > G1	
Sugary drinks, e.g., milk tea			
G1: <1 time/month	172	1.9 ± 2.4 *	0.7 ± 1.1
G2: 1–3 times/month	205	2.5 ± 2.6	0.7 ± 1.4
G3: 1–6 times/week	150	2.5 ± 2.7	0.5 ± 1.1
G4: >=1 time/day	35	2.2 ± 2.9	0.9 ± 1.4
		#: G2, G3 > G1	
Soft drinks			
G1: <1 time/month	311	2.3 ± 2.6	0.6 ± 1.2
G2: 1–3 times/month	175	2.2 ± 2.6	0.7 ± 1.3
G3: 1–6 times/week	64	2.3 ± 2.6	0.8 ± 1.4
G4: >=1 time/day	13	2.9 ± 3.1	0.5 ± 0.9

^a^ Values are mean ± standard deviation. * *p* < 0.05, ** *p* < 0.01, Kruskal–Wallis test. # Groups with significant difference by Dunn’s multiple comparison procedure.

**Table 3 ijerph-18-07027-t003:** Correlations between anthropometric measures, age, number of teeth, and caries scores.

Variables	Age	BMI-Z-Score	Deft	DMFT
Age	1 ^a^	0.04 ^a^	−0.54 ***	0.37 ***
BMI-z-score	0.04	1	−0.07	0.05
Number of primary teeth	−0.86 ***	−0.12 **	0.59 ***	−0.38 ***
Number of permanent teeth	0.89 ***	0.13 **	−0.60 ***	0.38 ***
Height (cm)	0.88 ***	0.23 ***	−0.51 ***	0.32 ***
Weight (kg)	0.72 ***	0.65 ***	−0.44 ***	0.29 ***
BMI (kg/m^2^)	0.36 ***	0.94 ***	−0.25 ***	0.17 ***
Body fat (%)	0.26 ***	0.87 ***	−0.19 ***	0.14 **
Waist circumference (cm)	0.39 ***	0.78 ***	−0.28 ***	0.15 ***
Upper arm circumference (cm)	0.47 ***	0.82 ***	−0.31 ***	0.22 ***
Triceps skinfold (mm)	0.25 ***	0.77 ***	−0.20 ***	0.12 **

^a^ Spearman rank correlation coefficient, *N* = 569. ** *p* < 0.01, *** *p* < 0.001.

**Table 4 ijerph-18-07027-t004:** The relationship of weight status with sociodemographic factors, caries experience, medical history, and food intake frequency among schoolchildren.

			Weight Status	
Variable	*N*	Normal/Underweight	Overweight	Obese
All children	569	361 (63.4) ^a^	105 (18.5)	103 (18.1)
Age group				
6–8 year	261	176 (67.4)	41 (15.7)	44 (16.9)
9–12 year	308	185 (60.1)	64 (20.8)	59 (19.2)
Gender				
Boys	283	161 (56.9)	46 (16.2)	76 (26.9) ^b,^***
Girls	286	200 (69.9)	59 (20.6)	27 (9.4)
Father’s education level				
Less than high school	64	33 (51.6)	13 (20.3)	18 (28.1) ^b,^*
High school	229	135 (59.0)	46 (20.1)	48 (20.9)
College and above	270	188 (69.6)	45 (16.6)	37 (13.7)
Mother’s education level				
Less than high school	51	25 (49.0)	9 (17.7)	17 (33.3) ^b,^**
High school	193	128 (66.3)	27 (14.0)	38 (19.7)
College and above	313	200 (63.9)	67 (21.4)	46 (14.7)
Frequency of tooth-brushing				
<=1 time/day	96	59 (61.5)	15 (15.6)	22 (22.9) ^b,^*
2 times/day	308	206 (66.9)	48 (15.6)	54 (17.5)
>=3 times/day	163	95 (58.3)	42 (25.8)	26 (15.9)
Caries experience				
Primary teeth				
Yes (deft ≥ 1)	359	243 (67.6)	61 (16.9)	55 (15.3) ^b,^*
No (deft = 0)	210	118 (56.1)	44 (20.9)	48 (22.8)
deft score	569	2.5 ± 2.7	2.1 ± 2.4	1.8 ± 2.3 ^c,^*
Permanent teeth				
Yes (DMFT ≥ 1)	180	110 (61.1)	38 (21.1)	32 (17.7)
No (DMFT = 0)	389	251 (64.5)	67 (17.2)	71 (18.2)
DMFT score	569	0.6 ± 1.2	0.8 ± 1.3	0.7 ± 1.2
Number of teeth				
Primary teeth	569	8.6 ± 6.0	7.1 ± 5.9	6.9 ± 5.6 ^c,^**
Permanent teeth	569	14.7 ± 7.3	17.1 ± 7.2	16.7 ± 6.8 ^c,^**
Medical history and activity				
History of allergic rhinitis				
Yes	119	75 (63.0)	23 (19.3)	21 (17.7)
No	441	280 (63.5)	81 (18.4)	80 (18.1)
Physical activity				
High	149	109 (73.2)	25 (16.8)	15 (10.0) ^b,^**
Moderate	345	214 (62.0)	66 (19.1)	65 (18.8)
Low	73	37 (50.7)	13 (17.8)	23 (31.5)
Food intake frequency				
Fruit juice				
<1 time/month	289	174 (60.2)	59 (20.4)	56 (19.3)
1–3 times/month	127	87 (68.5)	23 (18.1)	17 (13.3)
1–6 times/week	91	62 (68.1)	15 (16.4)	14 (15.3)
>=1 time/day	55	33 (60)	8 (14.5)	14 (25.4)
Yakult & other yogurt drinks				
<1 time/month	190	121 (63.6)	33 (17.3)	36 (18.9)
1–3 times/month	210	129 (61.4)	45 (21.4)	36 (17.1)
1–6 times/week	123	79 (64.2)	24 (19.5)	20 (16.2)
>=1 time/day	42	30 (71.4)	3 (7.14)	9 (21.4)
Fast food				
<1 time/month	145	96 (66.2)	24 (16.6)	25 (17.2)
1–3 times/month	265	166 (62.6)	51 (19.2)	48 (18.1)
1–6 times/week	133	83 (62.4)	25 (18.8)	25 (18.8)
>=1 time/day	20	12 (60.0)	4 (20.0)	4 (20.0)
Candy & chocolate				
<1 time/month	87	54 (62.1)	18 (20.7)	15 (17.2)
1–3 times/month	187	107 (57.2)	43 (23.0)	37 (19.8)
1–6 times/week	228	159 (69.7)	31 (13.6)	38 (16.7)
>=1 time/day	62	38 (61.3)	13 (21.0)	11 (17.7)
Cake, bread & cookies				
<1 time/month	55	31 (56.3)	10 (18.1)	14 (25.4)
1–3 times/month	130	77 (59.2)	29 (22.3)	24 (18.4)
1–6 times/week	288	187 (64.9)	51 (17.7)	50 (17.3)
>=1 time/day	92	63 (68.4)	15 (16.3)	14 (15.2)
Sugary drinks				
<1 time/month	172	114 (66.3)	35 (20.3)	23 (13.4)
1–3 times/month	205	133 (64.9)	31 (15.1)	41 (20.0)
1–6 times/week	150	86 (57.3)	35 (23.3)	29 (19.3)
>=1 time/day	35	24 (68.5)	3 (8.6)	8 (22.9)

^a^ Values are given as *n* (%) or mean ± standard deviation. ^b^ Chi-squared test. ^c^ Kruskal–Wallis test. * *p* < 0.05, ** *p* < 0.01, *** *p* < 0.001.

**Table 5 ijerph-18-07027-t005:** The association between weight status and caries scores, results from the negative binomial regression analyses.

Dependent Variable	Independent Variable	Unadjusted OR ^a^ (95% CI)	Adjusted OR ^b^ (95% CI)	Adjusted OR ^c^ (95% CI)
deft	Weight status			
	Normal/underweight	1	1	1
	Overweight	0.82 (0.62–1.08)	0.92 (0.72–1.17)	0.94 (0.74–1.20)
	Obese	0.70 (0.52–0.93) *	0.66 (0.51–0.85) **	0.81 (0.63–1.05)
DMFT	Weight status			
	Normal/underweight	1	1	1
	Overweight	1.40 (0.92–2.13)	1.38 (0.94–2.04)	1.30 (0.89–1.91)
	Obese	1.09 (0.70–1.68)	1.14 (0.76–1.72)	0.92 (0.61–1.39)

Abbreviations: OR, odds ratio; CI, confidence interval. ^a^ Without adjustment for covariates. ^b^ Adjustment for age, gender, father’s education level, mother’s education level. ^c^ Adjustment for gender, father’s education level, mother’s education level, and number of primary teeth (for deft) or permanent teeth (for DMFT). * *p* < 0.05, ** *p* < 0.01.

## Data Availability

The data analyzed and presented in this manuscript are available on request from the corresponding author.

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
