# Peer review of "Anthropometric Status, Diet, and Dental Caries among Schoolchildren"

_ijerph, 2021, doi:10.3390/ijerph18137027_

Round 1

Reviewer 1 Report

Dear authors,

I had the opportunity of revising the present manuscript regarding the correlation among Anthropometric status, diet and dental caries among school-children.

The study is interesting but needs a major revision.

I list here some suggestions:

  1. The title needs to be revised, why you repeat two times “Anthropometric status and dental caries” in the same title? What kind of study is this? Please change the title according to this points.
  2. Did you perform a sample size/power analysis?
  3. How did you check the data distribution? Please add it to the text.
  4. Demographic data of the sample should be moved from the results to the methods participants section.
  5. What does "text only" mean at the end of the food intake frequency paragraph?
  6. You wrote in the discussion “previous studies assessed the association among children of different ethnicity/culture groups or geographical areas, and these factors might have modified the caries-obesity relationship” please add references as for example “Nota A, Caruso S, Cantile T, Gatto R, Ingenito A, Tecco S, Ferrazzano GF.Biomed Res Int. 2019 Dec 23;2019:7981687.” that shows an association with mother’s nationality. Why these factors were not considered in your study?
  7. The reference style needs a revision.
  8. Tables should be included in the text.

Best Regards

Reviewer 2 Report

This is an interesting cross-sectional study about the association between obesity and diet and dental caries among 6-12-year-old children in Taiwan. The topic of the study is interesting and valuable, however the study has a number of drawbacks which ought to be tackled before it can be considered for publication.

Title: "Anthropometric" ??

Introduction: restructuring of the introduction would certainly improve it.

Obesity has been found to be associated with infrequent tooth brushing habits among 16-year olds (Virtanen et al. 2019). It is obvious that this will lead to oral health problems later in life. The second paragraph is focused on dental caries, but most of the studies are cross-sectional in nature and no causal conclusions can be drawn using this design.

The aims could be rephrased along with the title.

Methods: the description of the subjects and methods is not comprehensive enough for the reader to get a clear picture of the study. 

Please define the sampling method.

More information is needed for the reader to get a clear picture of the Taiwanese community and the sampling. Using this sampling method, what was the representativeness of the sample?

The survey: how were the questions developed? Any referenses? Reliability and validity? Please provide more information for the reader.

How was the "calibration practice" done?

Discussion: it is not possible to draw causal conclusions based on a cross-sectional study.

Overall the study is interesting and valuable, and includes lots of variables. However no significant associations were seen. The study has potential, but the report should be more focused and should be amended.

Please use the BMI information obtained in reporting. Inclusion of information of plaque and use of ICDAS would have given much more information among the present 6-12-year-olds. Consider to omit part of the variables to get a more focused report. The tables ought to be revised and should be inclusive.

Round 2

Reviewer 1 Report

All the comments were answered and suggestions performed by the authors.

Author Response

Thank you very much for helping us improve the manuscript.

Reviewer 2 Report

The manuscript has improved and the authors have responded to most of the previous comments/critics. Some minor issues remain:

Some of the numerical information is presented with two decimals f.ex. the age and SDs. This is not necessary. Please reconsider the same regarding other variables (dmft) throughout the manuscript. 

In the previous evaluation, revision of the tables was requested. The tables are not very clear to read and f.ex. some of the categories do not provide any valuable information and could be combined.

WHO criteria was used in this study, but f.ex. reported 3 times/day tooth brushing is presented. Does this add to the paper?

In the Conclusions (Abstract + Discussion) the age of the children should be added to schoolchildren.
